# Give Them Christ: Native Agency in the Evangelization of Puerto Rico, 1900 to 1917

**Angel Santiago-Vendrell**

Asbury Theological Seminary, Orlando, FL 32825, USA; a.santiago-vendrell@asburyseminary.edu

**Abstract:** The scholarship on the history of Protestant missions to Puerto Rico after the Spanish American War of 1898 emphasizes the Americanizing tendencies of the missionaries in the construction of the new Puerto Rican. There is no doubt that the main missionary motif during the 1890s was indeed civilization. Even though the Americanizing motif was part of the evangelistic efforts of some missionaries, new evidence shows that a minority of missionaries, among them Presbyterians James A. McAllister and Judson Underwood, had a clear vision of indigenization/contextualization for the emerging church based on language (Spanish) and culture (Puerto Rican). The spread of Christianity was successful not only because of the missionaries but also because native agents took up the task of evangelizing their own people; they were not passive spectators but active agents translating and processing the message of the gospel to fulfill their own people's needs based on their own individual cultural assumptions. This article problematizes the past divisions of such evangelizing activities between the history of Christianity, mission history, and theology by analyzing the native ministries of Adela Sousa (a Bible woman) and Miguel Martinez in light of the teachings of the American missionaries. The investigation claims that because of Puerto Rican agents' roles in the process of evangelization, a new fusion between the history of Christianity, mission history, and theology emerged as soon as new converts embraced and began to preach the gospel.

**Keywords:** Americanization; Presbyterian missionaries; local bible women; native evangelists





## 1. First Impressions of Presbyterian Missionaries in Puerto Rico

After the Spanish American War, on August 12, 1898, the Treaty of Paris ratified Puerto Rico as a United States territory. Within eighteen months, the US congress passed the Foraker Act, making Puerto Rico the first unincorporated territory of the United States (Denis 2015). In 1917, the congress of the United States passed the Jones Act that granted US citizenship to all Puerto Ricans on the island. Puerto Rico thus became the first colony of the United States, initiating a process by which the island became an object of the political, economic, cultural, and ideological apparatus of the United States. As soon as US troops invaded the island and began to develop a military government, missionaries were already planning the best way to evangelize the new territory.

On 13 July 1898, the Presbyterian Board of Foreign Missions invited other Protestant mission agencies and denominations to their main office in New York to discuss how to proceed in the evangelization of Puerto Rico (Walsh 2008). These missionary agencies came to an agreement on how to divide the island because they did not want to duplicate their work and also because they wanted to give to the people of Puerto Rico a unified evangelical testimony (Odell 1952). The Presbyterians, Baptists, Congregationalists, Disciples of Christ, Brethren, and Methodists were represented at the meeting. The general terms of the agreement were that:

The Congregationalists would develop the eastern part of the island, the Presbyterians took the western end, the Christian Alliance entered the town of Manati on the North, the United Brethren occupied several towns on the south shore, while the Baptists, Methodists, and Disciples of Christ placed their workers more towards the center of the whole area.

In San Juan, the largest city of the island, were the Episcopalians, Baptists, Lutherans, Presbyterians, Methodists, and the Young Men's Christian Association, while in Ponce, the southern metropolis, are the Disciples of Christ, United Brethren, Disciples of Christ, the Baptists, and the Protestant Episcopal Church (Committee on Cooperation in Latin America 1917).[1]

This article problematizes the past divisions between the history of Christianity, mission history, and theology by analyzing the native ministries of Adela Sousa (a Bible woman) and Miguel Martinez in light of the teachings of the American missionaries. The investigation claims that because of Puerto Rican agents' roles in the process of evangelization, a new fusion between the history of Christianity, mission history, and theology emerged as soon as new converts embraced and began to preach the gospel.

Before the invasion, the policy of the Roman Catholic Church and the Puerto Rican government did not allow Protestant missionaries to proselytize on the island. The missionaries took advantage of the new political situation by preaching the gospel freely everywhere they could, on street corners and public plazas, at house services, and to everyone they encountered in their path. They perceived Puerto Ricans as "a people long bound with chains of spiritual darkness, ignorance, and superstition" (Drury 1924). Other missionaries saw them as children in need of tutelage because for centuries they had been subjected to Spanish rule and Roman Catholicism (Groose 1910). The Presbyterian missionaries considered Roman Catholicism to be a false religion. For them, "Porto Rico is a country that have no Sabbath, no Bible, no preaching of the gospel, a very limited and imperfect educational system; a country destitute of the means of grace as the darkest part of Africa or the farthest community in China" (Halsey 1902). Perhaps the missionaries painted a bleak picture of Puerto Rico to the people back home in the United States with the intent to secure funds for their work. In any case, Presbyterian missionaries had a very negative view of the Spanish system of government and perceived Roman Catholicism as an aberration from the devil.

In terms of vices, Presbyterians considered Puerto Ricans to be immoral because of their lack of 'true' Christian teachings. Their gambling, rooster-fights, tavern-alcohol, and dancing missionaries interpreted as mortal sins. For example, Edwin A. McDonald, missionary to the town of Isabela, decried rooster-fighting and gambling near his preaching post as acts of cruelty against animals. When he confronted the men about their actions, he saw an almost blinded rooster, bloodied from combat, and close to death. Shocked and full of anger, he describes his reaction: he took "a man by the collar and opening my knife, made as though I would pick out his eyes, and asked them what would they think of me if I was strong enough to hold him and would deliberately cut his face and pick out his eyes for my amusement" (McDonald 1907, p. 200). After the confrontation, the men listened to McDonald speak about Christ and his love for animals as creatures of God.

The "sexual laxity" of Puerto Ricans similarly astonished other missionaries. According to Arnold Smith, a missionary to Añasco, adultery was a common thing among Puerto Ricans. He blamed the Roman Catholic priests for this, for two reasons: (1) the priests charged an exorbitant amount of money for weddings, and (2) the priests themselves participated in sexual immorality by having children with parishioners (Smith 1907). According to Judson Underwood, the immoral practice of concubinage should disappeared because the missionaries did not charge money for weddings (Underwood 1908). To substantia his claim, he noted that he had performed 130 weddings in fifteen months.

Perhaps the missionaries deplored the *fiesta patronal* or patron saint festivities the most because for them these amounted to one big celebration in which alcohol, smoking, gambling, and free sexual expression played crucial roles in corrupting Puerto Ricans. Missionaries likewise perceived the fiesta's street processions of the virgin and saints to be idolatrous. They considered the spirit of the *fiesta patronal* to be a deception by the "Romanists' to keep islanders under their spell. It is not clear from their writings what

---

1    (Committee on Cooperation in Latin America 1917).

precisely gave missionaries the impression that the *fiesta* was rife with sexual immorality, but Underwood reports that it led to an "annual orgy" ([Halsey 1902](#), p. 203). Milton Greene went so far as to claim that the *fiestas patronales*, "enshrouded in the midst of traditional sainthood and virginal intercession, taught a religion of penances, accustomed to the sight of selfish and sensual priestsm" and that they led the masses to their doom by fomenting "idolatry, indolence, sensualism, distrustfulness, and even bestiality" ([Greene 1900](#), p. 851). The missionaries countered such apparent behaviors by offering the people Christ and instilling in them love for their new colonial overlords.

The scholarly literature on Protestant efforts to Americanize Puerto Ricans is extensive.[2] Graeme S. Mount summarizes, "Presbyterians were enthusiastic, if unwitting, agents of empire in Puerto Rico. They were proud to be American, and in Americanizing Puerto Rico, they believed that they were doing God's will and providing a service to Puerto Rican people" ([Mount 1977](#), p. 242). Presbyterian missionaries interpreted the events of the Spanish–American War in terms of manifest destiny guided by the providence of God. For missionaries like Underwood, Puerto Rico and its people were already American because "God's providence had been placed under their protection" ([Underwood 1910](#), p. 207). This is understandable to a certain degree because the providence of God is one of the primary teachings of Reformed theology, to which Presbyterians adhered ([Lochmann 1997](#)). Such a theology insists that there is nothing in this world or the universe which God does not control. God's providence is in control of everything that occurs in heaven and earth. If God is in control of everything that happens in the universe, Presbyterians rationally interpreted the events of the Spanish American War by Presbyterians as being preordained by God before the foundation of the world. That being the case, Presbyterian missionaries were duty-bound to obey God by preaching the gospel and Americanizing the newly acquired territory.

Milton Greene wanted to bring to Puerto Ricans not only what he considered to be the blessings of American principles, methods, and laws, but especially the true gospel of Christ. For him, there were two ways in which this process of Americanization could happen: first, "By a cordial acceptance of American civilization, with its equal advantages, rights and responsibilities for all classes and conditions" ([Greene 1900](#), p. 849). He insisted that the masses in Puerto Rico should not listen to and follow nationalist leaders, even though the new government was a military regime, in which most positions of power were occupied by people from the United States. Second, if the Puerto Rican people wanted to follow the "socialists and demagogues," then "American principles, methods, institutions and laws [would have to] be introduced by American power. Cost what it may, Porto Rico is to be Americanized. If we must accept the conclusion that no argument but force will be appreciated, then force will doubtless be employed" (Ibid., p. 850).

Perhaps he had in mind the conflict between the United States and the Philippines. When the United States took possession of the Philippines, entering Manila on August 13, 1898, President McKinley pointed out,

When next I realized that the Philippines had dropped into our lap, I confess I did not know what to do with them. I sought counsel from all sides—Democrats as well as Republicans—but got little help ... I am not ashamed to tell you, gentlemen, that I went down on my knees and prayed Almighty God for light and guidance more than one night. And one night late it came to me this way ... (1) That we could not give them back to Spain—that would be cowardly and dishonorable; (2) That we could not turn them over to France or Germany—that will be bad business; (3) That we cannot leave them to themselves—they were unfit for self-government—and they will soon have anarchy and misrule over there worse than Spain was; and (4) That there was nothing left for us to do but to take them all, and to educate the Filipinos, and uplift and *civilize* and Christianize them ([Stuntz 1904](#), p. 144).

---

2 ([Walsh 2008](#); [Gotay 1997](#); [Mount 1977](#); [Urrego 2002](#)).

Not surprisingly, people in the Philippines did not receive this declaration by the President of the United States with open hearts. Indeed, it ignited the Filipino-American War, which lasted three years, longer than the war against Spain, with the Filipinos finally succumbing to the Americans only in May 23, 1901. For the missionaries, McKinley's words were a confirmation of their theology of the providence of God. To his mind, God had given the Philippines, Guam, and Puerto Rico to the United States and "the presence of the American flag is the pledge of American perpetuity" in these islands (Halsey 1902, p. 173).

Not only McKinley, but also Western missionaries considered Puerto Ricans to lack the means for self-government. This is a constant theme in Presbyterian missionary literature. The editorial of *The Assembly Herald* in 1902 described Puerto Ricans "[a]s a people with no experience in self-government and are practically ignorant of the meaning of the phrase." Six years later, Raymond Hildreth, a medical doctor in charge of the Presbyterian Hospital in San Juan likewise claimed, "Self-government is impossible for a people without years of training...We consider education and the public school to be fundamental necessities in self-government. But without personal religion, education, self-government and material prosperity will never suffice to transform Porto Rico into a great state of self-governing people" (Hildreth 1908, p. 204). Robert McLean, a missionary appointed in 1902 to San Juan, likewise pointed out, "The duty of government is to prepare them as rapidly as possible for self-government under our Constitution, and the burden of that preparation is too great for the island alone" (McLean 1908, p. 206).

Yet, the missionaries believed that the problems of Puerto Rico would not be solved by the government alone. Instead, they were convinced that their work of evangelizing the masses would create a new class of people who would become good American citizens. As Milton E. Caldwell, missionary appointed to the town of Mayagüez, pointed out, "A much higher purpose, however, than to make good citizens, should fill us and urge us to immediate action; namely, to bring these perishing and suffering multitudes to Christ and salvation" (Caldwell 1902, p. 175).

Yet, such missionaries had in common a belief that the evangelistic work in Puerto Rico should be conducted by natives, not Western missionaries. Underwood pointed out, "If the gospel is to be for Puerto Rico what it has been for other nations, the basis for a real and effective prosperity, it is necessary that there be a sufficient number of native workers to spread it, therefore we have begged the Lord of the harvest to send workers to his harvest" (Underwood 1908, p. 380). For this reason, Judson Underwood and James McAllister proposed that the Presbyterian mission should establish a theological seminary in Mayagüez.

Presbyterian missionaries were well educated. Underwood completed his degree from McCormick Theological Seminary in 1896 and McAllister graduated from Gettysburg College and Princeton Theological Seminary with additional studies at Princeton University and Columbia University (Odell 1952, p. 23). The Presbyterians opened Seminario Teológico de Mayaguez (STM) on October 1, 1906 in the town of Mayaguez. The United Brethren joined the Presbyterian training school in 1912, which changed its name to Seminario Teológico Puertoricense adding one faculty member to the previous three missionary teachers. The missionaries planned a course of study that would last six years of work—two years of preparatory studies for students with no prior education, and four years of combined college and seminary work. The seminary developed a winter and summer program designed for students to work with a missionary supervisor in the areas of preaching, evangelizing rural communities, and organizing Sunday schools (McAllister 1908, p. 207). In its first year of operation there were fifteen students registered. In its second year of operation, STM doubled the number of students to twenty-nine. Underwood pointed out, "Coming as these young men did, many of them raw boys from the country, they applied themselves to their new work of preparation for the Gospel work with marked success and a tenacity that is rare in these lands" (Underwood 1907c, p. 577).

The Survey on Education of Latin America at the Panama Conference of 1916 hailed the project a success story of theological collaboration. The Report stated that, "The

oldest effort in union theological seminaries in Latin America is the very successful school at Mayaguez, Puerto Rico, where the Presbyterians and United Brethren unite in a school which draws pupils from many communions in different parts of the West Indies" (Committee on Cooperation in Latin America 1917). Underwood and McAllister wanted the seminary to serve not only native Puerto Ricans but also prospects from all the West Indies and Latin America.

In 1914, James McAllister article "Un Ministerio Nativo Bien Preparado" in *Puerto Rico Evangélico* described what he meant by training a well-educated minister. First, ministers were the religious guides of people on pilgrimage to heaven. Second, the best agent to proclaim the gospel to her/his own people should be a native minister. Finally, native ministers were to be educated for the task of preaching the gospel. McAllister pointed out, "For ministers to be successful as spiritual guides of people they must be educated because they are the ones leading people in the midst of a world in need. A native minister without education would be a catastrophe. The educational norms for native ministers should be equal to the norms in the United States" (McAllister 1914).

## 2. Adela Sousa: A Dedicated Bible Woman

History is often written by the ones in power. The history written by Presbyterian missionaries often contained the term "a native helper." In most cases the missionaries did not reveal the gender of the "native helper," but in a few instances the native helper was recognized as a Bible woman. An example is the case of Adela Sousa, a Bible woman in the Aguadilla station who worked for the Presbyterian church under Judson Underwood. Underwood had served in Brazil for four years as a lay educational worker. He returned to the United States to complete his studies at McCormick Theological Seminary in 1896. He arrived in Puerto Rico in 1900, and in less than two years, the mission grew to two-hundred members. Among those two-hundred members there were three that he appointed as workers. In this early stage of the mission, the missionaries created their own denominational course of study, which typically entailed one missionary instructing one disciple close to him, thus opening a path for the ordination of the native leaders. Underwood appointed Sousa as a Bible woman to visit the sick, encourage the members of the church, and to minister to the "upper class women" in their homes. He said, "It is a delight to see the wonderful grasp she has on Scripture truth, and her wise, tactful and convincing manner of presenting the gospel. She not only teaches among the people, but teaches a class of sixty children in the Sabbath-school, [and] holds three-night classes weekly to teach adults to read and write" (Halsey 1902, p. 177).

In 1902, Underwood commissioned Adela Sousa to work in the town of Quebradillas. As the Bible woman, she visited the sick, encouraged converts in their homes, taught the Bible, and evangelized by visiting house-to-house. Four months after arriving in Quebradillas, Sousa had expanded the work to four new preaching posts in the villages of Los Cocos, Terra Nova, San Justo, and Capital. After a successful ministry in Quebradillas, Underwood appointed Sousa as an evangelist-at-large in the town of Isabela. A native worker named Jose Arroyo received Adela Sousa in the district with great enthusiasm. He praised Sousa as a sincere and fervent evangelist for the cause of the Lord and he asked for prayers for Sousa, that God might give her wisdom, courage, and power to proclaim the gospel (Arroyo 1908). Sousa tirelessly served the Lord in her work as a Bible woman. She had the trust of the missionaries and native leaders alike.

There were many other unnamed Bible women who worked in the evangelization process in Puerto Rico with the same zeal and success as Adela Sousa. For example, James Greer Woods, missionary in the San German station, narrated the story of two unnamed Bible women of La Pica church. These young women "with Testament in hand, made repeated visits to Sabana Grande, calling on the church members there and going from house to house among other families . . . They went about explaining or defending the Gospel and they accomplished an immense amount of good, especially among the women of the community" (Woods 1910, p. 120). Another unnamed woman in the same town

opened her house in the afternoons to be used as a school to teach poor boys and girls how to read and write, and every Sunday together with the missionary she conducted a service in her house (Lheureaux 1907). There were other women who, like the Samaritan woman in the Gospel of John, went back to their villages and invited their neighbors to hear the missionary preach. Underwood told the story of one such woman, "a devout widow" who lived six miles from San German in a rural area. The "devout widow" asked the new missionary to the region, E.S. Lheureaux, to visit her rural community. She gathered forty people the first Sunday. Just after three weeks, the new missionary post had more than one-hundred followers (Stuntz 1904). Victoriano Fernandez, a helper to Underwood and later the editor of *La Voz Evangélica*, toured the towns of Aguadilla, Moca, San Sebastian, Lares, and Isabela to present and propagate the weekly periodical. He noticed during his visit to San German that Angel Arroyo Rivera and Trinidad Dominguez were in charge of the work while Lheureaux attended the Presbyterian General Assembly. Fernandez identified Dominguez as a "powerful Bible woman" who "works tirelessly for the Lord." He also observed that in Aguadilla, two Bible women identified as Isabel Sales and Maria de la Tejera were "faithful servants of the Lord" (Fernandez 1908, p. 418).

According to Jose López, one of the first Puerto Ricans ordained to ministry, "the Bible woman was the most important agent in the missionary work to evangelize the island" (López 1907, p. 59). These Bible women were better known to their communities than the missionaries. They were more than mere helpers: they were missionaries and evangelists in their own right. The Bible woman thrived, building personal relationships by visiting homes in the villages. Due to their interpersonal relationships, Bible women conducted evangelism in an atmosphere of trust, respect, and love. It seems that in Puerto Rico these Bible women were affluent and well-educated because they ministered to upper-class women who were either disillusioned with Roman Catholicism or curious about Protestantism but did not dare to visit a Protestant service. Regardless of the population among whom they ministered, these Bible women were crucial in the development of Christianity in Puerto Rico. The other major group working to establish Christianity in Puerto Rico was native male workers, among them Miguel Martinez.

### 3. Miguel Martinez: Christ the Redeemer

Miguel Martinez was born in 1882 in the town of San German. He came from a wealthy family and wanted to be a science teacher before accepting Christ as savior. His conversion happened through a sermon preached by Jeff Woods at La Pica church in the town of San German. He was one of the first students at the Seminario Teologico de Mayagüez when it opened in 1906 and shortly thereafter appointed as a missionary helper. After four years, he graduated on 10 May 1910 and was ordained that same year in the town of Naranjito (Drury 1912). Seminary students were supposed to work in ministry during their summer and winter breaks. After his first year of studies, Martinez began to work with the Methodist church in Utuado during the summer of 1907. Utuado is in a rural area in the center of the island. Due to the comity agreement and the creation of the "Federation of Evangelical Churches in Puerto Rico" that began in 1905, Protestant missionaries had good relations among themselves and often collaborated in spreading the gospel through occasional jointly sponsored events. The objectives of the "Federación" were to show a unified witness, cultivate a spirit of fraternity among Protestant denominations, and cooperate as much as possible with other denominations to spread the gospel in Puerto Rico (Federación 1905). Martinez had a fruitful summer in Utuado working for the Methodists. He opened a preaching station and Sunday school in a village named Cuba, where he preached three times a week and taught the Sunday-school attended by fifty students (Underwood 1907b). He worked the rest of his seminary years as a traveling evangelist visiting the towns of San German, Añasco, Mayaguez, and Isabela (Underwood 1907a). In 1912, the missionaries appointed him as superintendent in charge of the work in Maricao y Las Marias (Drury 1913).

Martinez had great interest in missionary work. *La Voz Evangélica*, the Presbyterian periodical in Spanish, dedicated a page in every issue to world mission, similar to the one in *The Missionary Review of the World*. In the section "El Campo Misionero," Martinez contributed news from around the world. For example, the missionary news from China drew attention to the work of the Religious Tract Society of London, which had translated into Chinese and distributed 100,000 copies of the tracts The Traveler's Guide. In Japan, Count Okuma praised the missionaries and their Sunday services as uplifting Japanese culture. In India, philanthropist Ratan Tata donated six thousand dollars for a new building for Christian services. In the African Congo, the relationship between missionaries and locals were particularly fruitful and the influence of Christianity transformed people's lives. In Brazil, he narrated the story of Alexina Magallanes, a descendant of the Portuguese conquistador. She had converted to the gospel and served the Presbyterian church of Sao Joao del Rei as a Bible woman (Martinez 1913a).

Not only did Martinez write a weekly news column of missionary news, he also translated missionary biographies such as *Robert Moffat: Missionary to Africa* by M.E. Ritzmann, a sermon titled "The Un-rescued" by the prohibitionist, suffragist, and politician Marie C. Brehm, and Algernon J. Pollock's essay "Modern Spiritualism Briefly Tested by Scripture" (Martinez 1913b, p. 9). These three topics were crucial in the development of Protestantism in Puerto Rico, as the church became missionary by its very nature, defended prohibitionism, and challenged Spiritism through the gospel of Jesus Christ.

Even though Martinez did not equate the gospel with Americanization, he did understand Christianity as a civilizing force (Martinez 1907c). According to William Hutchison, "The common activistic premise of American liberals and revivalists—the point in which they were alike despite their differences—seemed in this era to be translating themselves into a paean of enthusiasm for the conquest of the world not only for Christ but for Christian civilization" (Hutchison 1982, p. 197). For Martinez, the technological advances of western civilization were a product of the gospel of Jesus Christ. The Roman Catholic Church accused Protestant missionaries of trying to erase Puerto Rican culture by introducing a foreign religion (Urrego 2002). Martinez countered this accusation by stating that Protestant missionaries in Puerto Rico lacked state support to impose their doctrines on the population. He pointed out, "This issue of Americanization is for the government, our duty is to evangelize, to spread the gospel to the whole island. We proclaimed Christ without asking those who received it if they are American or Anti-American. Christ wants all men without making any distinction because of political ideology" (Martinez 1907c, p. 119). Martinez believed that the gospel of Jesus Christ was the only redeeming path for Puerto Rico, that the day of salvation was at hand, and that no political alliance or process of Americanization would bring joy and peace to a sinful and depraved heart (Martinez 1908).

Martinez preached a gospel of redemption in which the giver of life came to the world to save the world through his incarnation, crucifixion, and resurrection. Even at the cusp of World War I, Martinez rejected the idea of equating the United States with the gospel. In spite of the involvement of the United States in the First War, Martinez denounced the war and made a clear difference between Christ and Americanization. He saw himself in the battlefield in a dream looking at all the ravages of war when suddenly absolute darkness overcame him. Just as Paul had an experience in Damascus with the light of Christ, Martinez had an epiphany of Christ in the midst of war in which Christ "told me that war is not the answer. The slogans of democracy and freedom were overcome by the words 'You should not kill'" (Martinez 1917c, pp. 5–6). The power of the gospel manifested itself in the weakness of a God who came to live and die to redeem humanity. Martinez did not serve and proclaim an American Christ constructed in the imagination and after the image of the missionaries, but rather the Christ of the Bible, the One who said: "I am the bread of life. Whoever comes to me will never go hungry, and whoever believes in me will never be thirsty" (Martinez 1917a, p. 207). For this reason, he said, the concept of Christian nations was a fallacy. Nations could not claim Christianity as a

national religion because it created nominal Christians and the true meaning and identity of Christianity could be overshadowed by the selfishness which characterized a person without Christ. Martinez pointed out, "The sad spectacle of war has shown us two important lessons: first, the false notion that we had about a Christian nation, and second, the great need of the world to embrace the true gospel of Christ which demands love of neighbor" (Martinez 1917c, p. 207).

Martinez preached a personal Christ who wanted to be in a new relationship with human beings contrary to the impersonal entities of Puerto Rican *espiritismo* (Martinez 1913b). He preached about the intimacy of following Jesus to a new life of joy based on the gratitude of God who "loved the world that he gave his one and only Son, that whoever believes in him shall not perish but have eternal life" (Martinez 1907b, p. 4). His message of the role of Christ in the family played a crucial role in reforming the home as the starting place to improve society (Martinez 1907a). In this sense, the message that Martinez proclaimed pointed to create a better society rallying against alcoholism as one of the vices destroying Puerto Rico at the time (Martinez 1917b).

## 4. Conclusions

The assumption that native Christians would follow the missionaries in their quest to Americanize Puerto Rico is more nuanced and complicated than previous research on the Americanization process of Puerto Rico has shown. The spread of Christianity by native agency is undisputed. New converts saw in the gospel of Jesus Christ a viable solution to their personal problems and the problems of the island. Bible women such as Adela Sousa were coparticipants with the missionaries in evangelizing the masses. Even though her story has not been part of the literature dedicated to the process of evangelism in Puerto Rico, she, like countless other Bible women, were the foundations of the work in the island. They were fearless agents whose only religious purpose was "to present the vicarious blood of Jesus Christ as the only means to obtain the remission of sins and enter a new life rooted in the gospel" (Irvine-Rivera 1908, p. 370).

The ministry of Miguel Martinez is another example that Puerto Ricans were not in the same pace as the missionaries when it came to issues of Americanization. Martinez never equated the gospel with Americanization. He believed in the God of the Bible who sent Jesus Christ to redeem the world from sin. His ministry as a local evangelist, pastor, district superintendent, and steady contributor to *La Voz Evangelica*, showed a man committed to the gospel and not to a political ideology. For him, all the technological, cultural, and social advances in Western civilization were byproducts of the wisdom of the gospel of Jesus Christ. His ministry showed the centrality of native agency in the spread and establishment of the gospel in Puerto Rico.

The relationship between the Bible, theology, hermeneutics, and practice has been part of the history of Christianity since its inception. The reception of the good news by local Christians happened in a dynamic process of growth in the Lord in which believers found in the Bible clues to face their human longings. Both Sousa and Martinez received their theological education under the tutelage of missionaries. Their basic theological training followed a devotional life in which reading the Bible became foundational for the Christian life. Even though their training was encapsulated in the Western/Reformed tradition of a moralistic, individualistic/soul-salvation, and substitutionary atonement, local agents interpreted the Bible based on their contextual reality. The way native leaders such as Sousa and Martinez constructed and shaped the message of the Bible, especially the story of Jesus, to reorder their own personal lives testified to the transforming power of God. While Sousa and Martinez undoubtedly still retain vestiges and major influences of the American missionaries who trained them, nonetheless, they were catalysts to a new generation of Puerto Rican converts dedicated to indigenizing the gospel.

Native Puerto Rican agency in the spread of Christianity problematizes past divisions between the history of Christianity, mission history, and theology. When does mission history become church history? With the first convert, with the establishment of the

presbytery in Puerto Rico in April 1902, or when the first native offered a theological argument to someone? This article has shown that because of Puerto Rican agents in the process of evangelization, the previous lines between the history of Christianity, mission history, and theology are less clear than previously occurred as soon as new converts embraced and propagated the gospel.

**Funding:** This research received no external funding.

**Institutional Review Board Statement:** The study was conducted according to the guidelines of the Declaration of Helsinki.

**Informed Consent Statement:** Not Applicable.

**Data Availability Statement:** Not Applicable.

**Conflicts of Interest:** The author declares there no conflict of interest.

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
