# Peer review of "Give Them Christ: Native Agency in the Evangelization of Puerto Rico, 1900 to 1917"

_religions, doi:10.3390/rel12030196_

Round 1
Reviewer 1 Report
The unfolding argument, the historiographical crucible, and the evidence used make this paper a compelling read. To understand that some missionary groups depended upon Puerto Ricans to carry out the work of the Church from the beginning does complicate traditional historiography. A comparison of Puerto Rico with Cuba might be in order for later thought and writing. Most historical work demonstrates a "nationalization" of missions post Cuban revolution. That such work had begun early in the 20th century prompts questions for future historiographical comparison. Kudos!
Please work to refrain from utilization of the passive voice in the construction of sentences, particularly the over-dependence on "was" in the presentation. Your writing sacrifices a clear reference to agency when passive verbs intentionally hide the author(S) of the action. I'm a stickler for the any form of the verb, to be, utilized in writing. Just drop "was" as much as possible and sharpen the word choice to strengthen the clarity.
Author Response
I rewrite all of the passive voice as requested by the reviewer.
Reviewer 2 Report
This is a well-researched and well-cited piece. The historical detail is rich. However, I feel that the thesis is not strong. First, it seems like the thesis is made explicit at the end but it would be good to see more of it at the beginning (if I had not read the abstract, I would not have known where you were going). Second, the conclusion ought not to introduce so much new information (e.g. unpacking Martinez's theology). I would bring some of that more up into the body of the paper. Third, I'm not sure what is meant by the last line: "a new fusion between the history of Christianity, mission history, and theology occurred". That statement feels like it ought to be important but it comes across as nebulous.
I also was concerned about your analysis of Sousa & Martinez as being native, unadulterated, and biblical. First, every person has cultural/political biases. That is not necessarily a bad thing, but it must be acknowledged. I would like to know what Sousa & Martinez's were, lest we paint them as some sort of pure native. Second, it would be fascinating to me to hear what a truly authentic Puerto Rican Christian theology looks/sounds like. That would be a really unique contribution to this chapter. Third, the way you describe Sousa & Martinez's theology doesn't sound Puerto Rican, it sounds Reformed/Western: individualistic, soul-salvation, substitutionary atonement, moralistic (e.g. teetotalism). I would expect a Puerto Rican Christianity (not just Catholic, but even Protestant) to be more collectivistic, concerned about social justice, etc. Fourth, it seems inconceivable to me that Sousa & Martinez would not have been influenced by Western missionaries, if that's how they were trained. Sousa, for example, being a colporteur, was following in the vein of American "Bible women" in their itinerancy and evangelism.
All this is to say, nuancing that, while Sousa & Martinez undoubtedly still retained vestiges (or even major influences) from their American theological & cultural overseers, nonetheless they were the bridge or catalyst to a new generation of indigenization--this would come across as a more credible argument (rather than just simply saying that American missionaries were all colonizers, and native missionaries managed to throw off those imperialistic shackles and immediately got it right).
Author Response
Thanks for a thorough review. I adopted all your recommendations. peace
1) I improved the thesis statement and put it at the beginning of the article.
2) I moved the new information of Martinez from the conclusion to the body of the paper.
3) I rephrase the last sentence of the paper to make it less ambiguous.
4) At this early stage a truly Puerto Rican theology is not possible.
5) The reviewer is right about the Reformed/Western theology because of point 4.
6) I never said that Sousa and Martinez were not influenced by the missionaries. As the reviewer said, “it seems inconceivable to me that Sousa & Martinez would not have been influenced by Western missionaries.” I agree with the statement. I made the missionaries' influence more explicit if it was not clear before.
7) I adopted the argument that Sousa and Martinez were bridge or catalyst figures in the first stages of indigenization in Puerto Rico.